# Identification of Cell-Surface Proteins Endocytosed by Human Brain Microvascular Endothelial Cells In Vitro

**DOI:** 10.3390/pharmaceutics12060579

**Published:** 2020-06-23

**Authors:** Shingo Ito, Mariko Oishi, Seiryo Ogata, Tatsuki Uemura, Pierre-Olivier Couraud, Takeshi Masuda, Sumio Ohtsuki

**Affiliations:** 1Department of Pharmaceutical Microbiology, Faculty of Life Sciences, Kumamoto University, 5-1 Oe-honmachi, Chuo-ku, Kumamoto 862-0973, Japan; ishingo@kumamoto-u.ac.jp (S.I.); tmasuda@kumamoto-u.ac.jp (T.M.); 2Department of Pharmaceutical Microbiology, School of Pharmacy, Kumamoto University, 5-1 Oe-honmachi, Chuo-ku, Kumamoto 862-0973, Japan; 134p1011@st.kumamoto-u.ac.jp; 3Department of Pharmaceutical Microbiology, Graduate School of Pharmaceutical Sciences, Kumamoto University, 5-1 Oe-honmachi, Chuo-ku, Kumamoto 862-0973, Japan; 190y2006@st.kumamoto-u.ac.jp (S.O.); tuemura@keio.jp (T.U.); 4Institut Cochin, Universite de Paris, Inserm U1016, CNRS UMR8104, 22 rue Méchain, 75014 Paris, France; pierre-olivier.couraud@inserm.fr

**Keywords:** blood–brain barrier, cell-surface biotinylation, internalization, podocalyxin, proteomics

## Abstract

Cell-surface proteins that can endocytose into brain microvascular endothelial cells serve as promising candidates for receptor-mediated transcytosis across the blood–brain barrier (BBB). Here, we comprehensively screened endocytic cell-surface proteins in hCMEC/D3 cells, a model of human brain microvascular endothelial cells, using surface biotinylation methodology and sequential window acquisition of all theoretical fragment-ion spectra-mass spectrometry (SWATH-MS)-based quantitative proteomics. Using this method, we identified 125 endocytic cell-surface proteins from hCMEC/D3 cells. Of these, 34 cell-surface proteins were selectively internalized into human brain microvascular endothelial cells, but not into human umbilical vein endothelial cells (HUVECs), a model of human peripheral microvascular endothelial cells. Two cell-surface proteins, intercellular adhesion molecule-1 (ICAM1) and podocalyxin (PODXL), were identified as BBB-localized endocytic cell-surface proteins in humans, using open mRNA and protein databases. Immunohistochemical evaluation confirmed PODXL expression in the plasma membrane of hCMEC/D3 cells and revealed that anti-PODXL antibody-labeled cell-surface PODXL internalized into hCMEC/D3 cells. Immunohistochemistry further revealed that PODXL is localized at the luminal side of human brain microvessels, supporting its potential suitability for translational applications. In conclusion, our findings highlight novel endocytic cell-surface proteins capable of internalizing into human brain microvascular endothelial cells. ICAM1 or PODXL targeted antibody or ligand-labeled biopharmaceuticals and nanocarriers may provide effective targeted delivery to the brain across the BBB for the treatment of central nervous system (CNS) diseases.

## 1. Introduction

The blood–brain barrier (BBB) constitutes the dynamic interface between the systemic circulation and the brain parenchyma. The BBB selectively supplies macromolecules such as transferrin and insulin to the brain by receptor-mediated transcytosis (RMT), an endogenous system for macromolecule transport across the cell via receptor endocytosis, membrane trafficking, and exocytosis. Transferrin receptor (TFRC) functions as a major RMT receptor at the luminal side of the BBB [1]. Thus, antibodies against TFRC have been investigated for their ability to afford biopharmaceutical delivery to the brain [2]. Recently, an anti-human TFRC antibody supported the successful delivery of recombinant iduronate-2-sulfatase protein to the brain for the treatment of mucopolysaccharidosis II in an animal model and in humans [3,4]. However, the diverse TFRC expression in the peripheral tissues causes anti-TFRC antibodies to have a short half-life and low tissue selectivity, and raises safety concerns [5,6]. Insulin receptor (INSR) and low-density lipoprotein receptor-related protein 1 (LRP1) are alternative RMT receptors at the BBB [7,8]. However, a recent report revealed that antibodies against murine INSR and LRP1 did not cross the murine BBB [9]. Thus, the delivery of biopharmaceuticals to the brain across the BBB remains a major challenge in central nervous system (CNS) drug development.

Endocytosis of cell-surface proteins is the first step to initiate RMT. Recent reports demonstrate that highly expressed plasma membrane transporter proteins such as glucose transporter 1 (GLUT1, also known as SLC2A1) along with 4F2 cell-surface antigens (4F2hc, CD98, SLC3A2) at the human BBB serve as target molecules for brain delivery via endocytosis [9,10]. In addition, CD46 plays a role in the uptake of exosomes derived from a brain-metastatic cancer cell line at the human BBB [11]. Thus, both traditional receptors and cell-surface proteins that are endocytosed into human brain microvascular endothelial cells represent candidate RMT molecules for brain delivery across the BBB.

We and others have reported that cell-surface biotinylation assays allow the investigation of cell-surface protein endocytosis [12,13,14,15]. Sequential window acquisition of all theoretical fragment-ion spectra-mass spectrometry (SWATH-MS)-based quantitative proteomics is a powerful label-free method for the comprehensive investigation of protein expression levels in biological samples [16]. Our previous quantitative proteome analyses revealed that traditional RMT receptors such as TFRC, INSR, and LRP1, and endocytic SLC transporters such as GLUT1 and 4F2hc, were expressed in the human BBB model cell line hCMEC/D3 [17,18]. The hCMEC/D3 cell line is a model of human brain microvascular endothelial cells involved in the transport of endogenous and exogenous compounds across the BBB. Therefore, combining cell-surface biotinylation assays and SWATH-MS-based quantification proteomics enables the comprehensive identification of novel endocytic cell-surface proteins in brain microvascular endothelial cells.

The purpose of the present study was to comprehensively identify cell-surface proteins selectively internalized into hCMEC/D3 cells, using a cell-surface protein biotinylation assay and quantitative proteomics with SWATH-MS. In addition, we identified BBB-selective internalized cell-surface proteins using open databases of transcriptome, proteome, and immunohistochemistry data [17,19,20,21].

## 2. Materials and Methods

### 2.1. Cell Culture

hCMEC/D3 cells were established from human brain microvascular endothelial cells immortalized by transduction using hTERT and SV40 large T antigen [22]. Human umbilical vein endothelial cells (HUVECs) were freshly isolated from pooled donors (PromoCell, Heidelberg, Germany). hCMEC/D3 cells (passages 33–40) were seeded on collagen type I-coated 100 mm culture dishes (Corning, Armonk, NY, USA), and the cells were cultured in EBM-2 medium (Lonza, Basel, Switzerland) supplemented with 5% fetal bovine serum (FBS; BD Bioscience, San Jose, CA, USA), 5 mg/L ascorbic acid (Sigma-Aldrich, St. Louis, MO, USA), 1.4 mM hydrocortisone (Sigma-Aldrich), 10 mM LiCl (Merck, Frankfurt, Germany), 10 mM HEPES (Dojindo, Kumamoto, Japan), 1% chemically defined lipid concentrate (Thermo Fisher Scientific, Waltham, MA, USA), and 1% penicillin-streptomycin (Thermo Fisher Scientific) at 37 °C in a humidified 5% CO_2_ chamber. HUVECs (passages 3–5) were seeded on collagen type I-coated 100 mm culture dishes and cultured using the Endothelial Cell Growth Medium Kit (PromoCell) at 37 °C in a humidified 5% CO_2_ chamber.

### 2.2. Internalization Assay

The internalization study was performed following the procedure shown in Section 3.1. The cells were cultured on 100 mm collagen type I-coated dishes (approximately 2 × 10^6^ cells/dish) or 35 mm collagen type I-coated glass-bottom dishes (approximately 1 × 10^4^ cells; Matsunami Glass Ind., Ltd., Osaka, Japan) under each culture condition. The cells were washed twice with ice-cold phosphate-buffered saline (PBS) (137 mM NaCl, 2.7 mM KCl, 10 mM Na_2_HPO_4_-12H_2_O, 1.5 mM KH_2_PO_4_) on ice, and then incubated with 0.4 mM EZ-Link Sulfo-NHS-SS-Biotin (Pierce, Waltham, MA, USA) in PBS at 4 °C for 30 min with gentle shaking. Residual biotin was quenched by incubation with 100 mM glycine in PBS for 5 min three times on ice, and then the cells were washed once with ice-cold PBS (“Cell-surface” fraction). The cells incubated with PBS only were used as the “Control” fraction. 

To allow endocytosis, surface-biotinylated cells were washed three times with warmed PBS, then incubated with 20% human blood serum (Cosmo Bio, Tokyo, Japan) in PBS at 37 °C for 5 min. The cells were washed with ice-cold PBS three times to immediately arrest endocytosis. To remove residual biotinylated cell-surface proteins, the cells were incubated with ice-cold 2-mercaptoethane sulfonate Na (MESNA; MP Biomedicals, Santa Ana, CA, USA) buffer (50 mM MESNA, 150 mM NaCl, 1.0 mM EDTA, 0.2% bovine serum albumin, 20 mM Tris, pH 8.6) for 20 min at 4 °C, then washed twice with PBS (“Internalization” fraction). 

To examine residual surface-biotinylated proteins following incubation with MESNA buffer, the cell-surface biotinylated cells were incubated with cold PBS at 4 °C for 5 min, then treated with MESNA buffer following the same procedure used for the Internalization fraction (“Stripping” fraction). To recover internalized cell-surface proteins, the cells in each fraction were collected from the 100 mm dish to a low-protein-binding 1.5 mL tube using a scraper, then the tubes were centrifuged at 10,000× *g* for 1 min. After removing the supernatant, the cell pellets were resuspended with 0.5 mL of ice-cold RIPA buffer (Pierce) containing protease inhibitor (Sigma-Aldrich), and then lysed by sonication using a bath sonicator (AU-12C, Aiwa, Tokyo, Japan) (4 sonication cycles of 5 min each). The cell lysates of each fraction were centrifuged at 10,000× *g* for 10 min at 4 °C, and the supernatants were collected into new low-protein-binding 1.5 mL tubes. 

Proteins were collected using streptavidin magnetic beads (Thermo Fisher Scientific). After washing the magnetic beads following the manufacturer’s protocol, the beads were added into the cell lysates in 1.5 mL tubes. The tubes were then incubated at room temperature for 1 h with frequent tapping. The beads were collected using a magnetic plate and the supernatant was discarded. Magnetic beads bearing the biotinylated protein were washed three times with 300 μL of RIPA buffer and three times with 300 μL of 0.5 M NaCl in RIPA buffer. The beads were then extensively washed with 100 μL of Phase Transfer Surfactant (PTS) buffer (12 mM sodium deoxycholate, 12 mM *N*-lauroylsarcosinate, and 100 mM Tris-HCl (pH 9.0)). Elution was achieved by reduction in 100 μL dithiothreitol (DTT; 50 mM) in PTS buffer at room temperature for 1 h.

### 2.3. Fluorescence Microscopy

Following the internalization assay, the cells were fixed with 4% paraformaldehyde (PFA) in PBS for 10 min, treated with permeabilization buffer (90% methanol, 5% acetic acid), and then blocked with 10% FBS in PBS for 20 min. The cells were incubated with FITC-labeled streptavidin (ab136201; Abcam, Cambridge, UK) in 10% FBS in PBS for 1 h at room temperature. After washing three times with PBS, the cells were incubated with DAPI for 5 min and mounted using VECTASHIELD Mounting Medium with DAPI (H-1200; Vector Laboratories, Burlingame, CA, USA). Images were acquired using a BZ-X700 fluorescence microscope (KEYENCE, Osaka, Japan). Image processing was performed using Adobe Photoshop CS2.

### 2.4. Silver Staining

Eluted samples were incubated with sodium dodecyl sulfate-polyacrylamide gel electrophoresis (SDS-PAGE) sample buffer solution (Wako, Osaka, Japan), then separated via 5–20% gradient SDS-PAGE. The bands were visualized using a silver staining kit according to the manufacturer’s protocol (Wako), and images were obtained on an Omega Lum G imager (Aplegen, San Francisco, CA, USA).

### 2.5. Quantitative Proteome Analysis

The proteins in each fraction of the cells in the internalization study were digested with trypsin using the PTS method [23], and the trypsin-digested samples were analyzed by SWATH-MS on a 5600 TripleTOF instrument (SCIEX, Framingham, MA, USA) interfaced to the DIONEX Ultimate 3000 RSLC nano system (Thermo Fisher Scientific), as previously described [24]. The MS/MS data obtained via information-dependent acquisition, using one peptide aliquot per group, were analyzed using the ProteinPilot Software version 4.5 (SCIEX) with the Paragon algorithm and UniProt human reference proteome database for protein identification (https://www.uniprot.org/proteomes/UP000005640). Targeted peptide peaks were extracted from the SWATH data by PeakView Software version 2.1 (SCIEX) using the identified protein data from ProteinPilot, and the sum of the area values of specific peptide peaks from each protein was calculated as the protein expression level.

### 2.6. Identification of Internalized Proteins

The identified proteins in each fraction were annotated by GO analysis using the Database for Annotation, Visualization, and Integrated Discovery (DAVID) version 6.8 (http://david.abcc.ncifcrf.gov/home.jsp). The peak area ratio of proteins (“Cell-surface” fraction to “Control” fraction) was calculated, and the proteins were grouped on a binary logarithmic scale. The percentage of the proteins annotated with “Plasma membrane”, “Cell surface”, and “Cytoplasm” in each group was calculated. The threshold that excluded cytosolic proteins was determined according to the percentage of proteins annotated with “Cytoplasm” in each cell, and proteins exceeding the ratio threshold were defined as cell surface-expressed proteins. Stripping efficacy was calculated using the following formula:Stripping efficacy (%) = [1 − (protein expression level in “Stripping” fraction/protein expression level in “Cell-surface” fraction)] × 100

Biotinylated endocytic proteins were selected from among the “biotinylated cell-surface proteins” using two criteria. Criterion 1 consisted of a peak area ratio of “Internalization”-to-“Cell-surface” proteins ≤2.0, stripping efficacy >85%, and a peak area ratio of “Internalization”-to-“Stripping” proteins >2.0. Criterion 2 required a peak area ratio of “Internalization”-to-“Cell-surface” proteins >2.0, stripping efficacy >50%, and a peak area ratio of “Internalization”-to-“Stripping” proteins >2.0.

### 2.7. Immunohistochemistry

hCMEC/D3 cells cultured on collagen type I-coated slide glass were fixed with 4% PFA/PBS for 10 min. Following treatment with 1% Triton-X100 for 10 min, the cells were incubated in blocking solution (Nacalai Tesque, Tokyo, Japan) for 1 h at room temperature, incubated with the anti-PODXL antibody (ab150358, Abcam) overnight at 4 °C, washed with PBS containing 0.1% Tween 20, and incubated with the secondary antibody (goat anti-rabbit IgG H&L; ab97075, Abcam) for 1 h at room temperature. The cells were then washed with PBS containing 0.1% Tween 20 and mounted using VECTASHIELD Mounting Medium with DAPI (Vector Laboratories).

Paraffin sections of human cerebral cortex were purchased from BioChain Institute, Inc. (Hayward, CA, USA). Following deparaffinization, the sections were washed with PBS and heated for 20 min at 90 °C with antigen activation solution (HistoVT One, Nacalai Tesque,) for antigen activation. The sections were then incubated in blocking solution (PBS containing 0.3% Triton X-100, 0.1% bovine serum albumin, and 2% donkey or goat serum) for 2 h at room temperature. Sections were incubated with the anti-PODXL antibody (ab150358, Abcam) overnight at 4 °C, washed with PBS contains 0.1% Tween 20, and incubated with the secondary antibody (goat anti-rabbit IgG H&L; ab97075, Abcam) for 2 h at room temperature. Sections were washed with PBS containing 0.1% Tween 20 and mounted using VECTASHIELD Mounting Medium with DAPI (Vector Laboratories). To visualize brain microvessels, we used fluorescein-conjugated *Lycopersicon esculentum* lectin (FL-lectin, FL-1171, Vector Laboratories). Images were acquired using an FV3000 confocal laser microscope (Olympus, Tokyo, Japan) with diode lasers (405, 488, and 561 nm) as the excitation source, and using FLUOVIEW FV3000 software (Olympus). The images were taken in sequential scan mode (1–4 stacks/image). Image processing was performed using Adobe Photoshop CS2.

### 2.8. Internalization of Antibody-Labeled Cell-Surface Protein in the Cells

The anti-PODXL antibody (MBL, Nagoya, Japan) and its IgG isotype (MBL) were labeled with fluorescein (FL) using the Fluorescein Labeling Kit (Dojindo). hCMEC/D3 cells cultured on BioCoat Collagen I Culture Slide (Corning Life Sciences, Corning, NY, USA) were treated with FL-labeled anti-PODXL antibody or FL-labeled IgG Isotype for 30 min at 4 °C. After washing the cells with PBS, the cells were incubated at 37 °C for 5 min, then fixed with 4% PFA/PBS for 10 min, washed with PBS containing 0.1% Tween 20, and mounted with VECTASHIELD Mounting Medium with DAPI (Vector Laboratories). Images were acquired using an FV3000 confocal microscope (Olympus) and image processing was performed using Adobe Photoshop CS2.

### 2.9. Statistical Analysis

Three biological replicates were used in the SWATH-MS-based quantitative proteome analysis, and the data are expressed as means ± standard deviations (SD).

## 3. Results

### 3.1. Detection of Biotinylated Proteins in hCMEC/D3 Cells and HUVECs

hCMEC/D3 cells were used as a model of human brain microvascular endothelial cells, and HUVECs were used as a model of peripheral microvascular endothelial cells. The workflow of the identification of “biotinylated endocytic cell-surface proteins” in hCMEC/D3 cells and HUVECs is shown in Figure 1.

The biotinylation of cell-surface proteins and their internalization were examined using fluorescence microscopy (Labeling step, Figure 1). After treatment with sulfo-NHS-SS-Biotin for 30 min at 4 °C (Cell-surface fraction), fluorescence derived from FITC-labeled streptavidin was observed on the cell-surface of hCMEC/D3 cells and HUVECs. In contrast, after treatment with PBS for 30 min at 4 °C (Control fraction), no fluorescence derived from FITC-labeled streptavidin was detected in hCMEC/D3 cells and HUVECs (Figure 2a). To internalize the biotinylated cell-surface proteins into the cells, the cell-surface biotinylated cells were treated with 20% human serum for 5 min at 37 °C, and then the cells were washed with MESNA buffer to remove residual biotin on the cell surface (Internalization fraction). Following this treatment, fluorescence derived from FITC-labeled streptavidin was detected as dots in the intracellular space of hCMEC/D3 cells and HUVECs (Figure 2a). In contrast, without the internalization step (Stripping fraction), no fluorescence derived from FITC-labeled streptavidin was detected within hCMEC/D3 cells and HUVECs (Figure 2a).

The eluted proteins recovered using streptavidin magnetic beads were examined by silver staining (Purification step, Figure 2b), which revealed that bands of a variety of molecular sizes could be detected in all fractions of hCMEC/D3 cells and HUVECs (Figure 2b). The numbers of bands in the ”Cell-surface” fractions were greater than those in the “Control” fractions of hCMEC/D3 cells and HUVECs (Figure 2b), and the numbers of bands in the “Internalization” fractions were greater than those in the “Stripping” fractions (Figure 2b). These results indicate that our established method enables the identification of endocytic cell-surface proteins in hCMEC/D3 cells and HUVECs.

### 3.2. Identification of “Biotinylated Endocytic Cell-Surface Proteins” in hCMEC/D3 Cells and HUVECs

Silver staining suggested that the “Internalization” fractions had been contaminated with some non-biotinylated proteins and non-cell-surface proteins (black arrow, Figure 2b) because some proteins were non-specifically bound to streptavidin magnetic beads. To identify “biotinylated endocytic cell-surface proteins”, we established an identification method based on the peak area as measured by SWATH-MS. First, “biotinylated endocytic cell-surface proteins” in hCMEC/D3 cells and HUVECs were identified. The identification and measurement of the peak area of proteins in the “Control”, “Cell-surface”, “Stripping”, and “Internalization” fractions of hCMEC/D3 cells and HUVECs were performed (Identification step). The total numbers of identified proteins from all fractions of hCMEC/D3 cells and HUVECs were 563 and 399, respectively (Appendix A). The non-biotinylated proteins and non-cell-surface proteins (such as cytosolic proteins), which were not targeted for RMT, were eliminated using our threshold method. We excluded non-targeted proteins using our novel method (Data analysis step, Figure 1). The peak area ratio of the “Cell-surface”-to-“Control” fractions for each identified protein was calculated from the sum of the peak area of the unique peptides of each identified protein in hCMEC/D3 cells and HUVECs. The peak area ratios of the “Cell-surface”-to-“Control” fractions were grouped on a binary logarithmic scale in Venn diagrams (Figure 3a,b). The result indicated that 519 (92.2%) and 300 (75.2%) of all identified proteins in hCMEC/D3 cells and HUVECs, respectively, had “Cell-surface”-to-“Control” peak area ratios > 2.

All identified proteins were then grouped using the GO terms “Plasma membrane”, “Cell surface”, or “Cytoplasm” [25], and the percentage of all proteins annotated with each term were calculated in each group and summarized in Figure 3c and d. The proteins annotated with “Cytoplasm” were not identified as exhibiting “Cell-Surface”-to-“Control” peak area ratios > 8 in hCMEC/D3 cells or >4 in HUVECs (Figure 3c,d) whereas the proteins annotated with “Plasma membrane” or “Cell surface” tended to have “Cell-Surface”-to-“Control” peak area ratios > 8 in hCMEC/D3 cells and peak area ratios >4 in HUVECs (Figure 3c,d). Thus, the threshold for extraction of “biotinylated cell-surface proteins” from among all identified proteins was determined to be a “Cell-Surface”-to-“Control” peak area ratio of 8 in hCMEC/D3 cells and 4 in HUVECs, respectively. The numbers of proteins presenting a ratio greater than the threshold were 378 (67.1%) and 225 (56.4%) in hCMEC/D3 cells and HUVECs, respectively (Appendix A). These proteins were defined as “biotinylated cell-surface proteins” in hCMEC/D3 cells and HUVECs, and are listed in Appendix A. By contrast, in hCMEC/D3 cells and HUVECs, 185 and 174 of the identified proteins, respectively, were excluded as non-biotinylated cell-surface proteins (Appendix A). The “Cell-surface”-to-“Control” peak area ratio > 2 criterion, which excludes non-specific binding to streptavidin beads and tube walls, was not satisfied for 44 of the 185 identified proteins (in hCMEC/D3 cells), and for 99 of the 174 identified proteins (in HUVECs). The remaining 141 proteins (in hCMEC/D3 cells) and 75 proteins (in HUVECs) did not meet the “Cell-Surface”-to-“Control” peak area ratio threshold (>8 and 4, respectively), which excludes cytosolic proteins.

“Biotinylated endocytic cell-surface proteins” were selected from among the “biotinylated cell-surface proteins” by satisfying either Criterion A or B based on Stripping efficacy, the fold change of the “Surface”-to-“Internalization” fractions, and the fold change of the “Internalization”-to-“Surface” fractions; an example of each set of criteria is shown in Figure 3e. Among the “biotinylated cell-surface proteins”, 125 proteins (Criterion A: 119; Criterion B: 6) and 113 proteins (Criterion A: 104; Criterion B: 9) satisfied either Criterion A or B in hCMEC/D3 cells and HUVECs, respectively (Figure 3f,g, Appendix A). These proteins were defined as “biotinylated endocytic cell-surface proteins” in hCMEC/D3 cells and HUVECs, and are listed in Appendix A.

The top 10 highest peak areas of “biotinylated cell-surface proteins” and “biotinylated endocytic cell-surface proteins” in hCMEC/D3 cells and HUVECs are summarized in Figure 4. The highest peak areas of “biotinylated cell-surface proteins” in the hCMEC/D3 cells and HUVECs were observed for galectin-1 (LGALS1) and MUC18 (MCAM), respectively (Figure 4a,b). Comparison of all top 10 highest expressed proteins in hCMEC/D3 cells and/or HUVECs (total 16 proteins) showed that LGALS1 and MCAM commonly exhibited high peak areas of “biotinylated cell-surface proteins” and “biotinylated endocytic cell-surface proteins” in hCMEC/D3 cells and HUVECs (Figure 4c). In contrast, intercellular adhesion molecule 1 (ICAM1) and chondroitin sulfate proteoglycan 4 (CSPG4) were identified as “biotinylated cell-surface proteins” in hCMEC/D3 cells but not in HUVECs (Figure 4c), and keratin, type I cytoskeletal 10 (KRT10) was identified as a “biotinylated cell-surface protein” in HUVECs but not in hCMEC/D3 cells (Figure 4c).

The highest peak areas of “biotinylated endocytic cell-surface proteins” in the hCMEC/D3 cells and HUVECs were observed for TFRC and cell-surface glycoprotein and integrin β1 (ITGB1), respectively (Figure 4d,e). Comparison of all top 10 highest expressed proteins in hCMEC/D3 cells and/or HUVECs (total 15 proteins) showed that ITGB1 commonly exhibited a high peak area among “biotinylated endocytic cell-surface proteins” in hCMEC/D3 cells and HUVECs. In contrast, TFRC and ICAM1 were identified among “biotinylated endocytic cell-surface proteins” of hCMEC/D3 cells but not HUVECs (Figure 4f), whereas Cadherin-5 (CDH5) and ICAM2 were identified as “biotinylated endocytic cell-surface proteins” in HUVECs but not in hCMEC/D3 cells (Figure 4f).

### 3.3. Identification of Selected Endocytic Cell-Surface Proteins in hCMEC/D3 Cells and HUVECs

As biopharmaceuticals are primarily administered by intravenous injection, it is necessary to minimalize the internalization of biopharmaceuticals by peripheral microvascular endothelial cells for efficient delivery across the BBB. To comprehensively identify biotinylated cell-surface proteins selectively endocytosed by human brain microvascular endothelial cells, as opposed to those selectively endocytosed by peripheral microvascular endothelial cells, we selected “biotinylated endocytic cell-surface proteins” from among the “biotinylated cell-surface proteins” using four criteria (Figure 5a). First, we extracted the “biotinylated cell-surface proteins” and “biotinylated endocytic cell-surface proteins” that were identified in hCMEC/D3 cells but not in HUVECs (Criterion I). As a result, 214 proteins were identified as hCMEC/D3-selective “biotinylated cell-surface proteins, by subtracting the “biotinylated cell-surface proteins” found in both hCMEC/D3 cells and HUVECs (164 proteins, Appendix A) from “biotinylated cell-surface protein” in hCMEC/D3 cells (378 proteins, Figure 3f, Appendix A). Further, 61 proteins (“biotinylated endocytic cell-surface protein” in hCMEC/D3 cells, Appendix A) were identified as hCMEC/D3-selective “biotinylated endocytic cell-surface proteins”, by subtracting the “biotinylated endocytic cell-surface proteins” found in both hCMEC/D3 cells and HUVECs (64 proteins, Appendix A) from “biotinylated endocytic cell-surface proteins” in hCMEC/D3 cells (125 proteins, Figure 3f, Appendix A). Among the 214 “biotinylated cell-surface proteins” in hCMEC/D3 cells (open circle in the upper Venn diagram of Criterion I, Figure 5a), 49 proteins were identified as both “biotinylated endocytic cell-surface proteins” and “biotinylated endocytic cell-surface proteins” in hCMEC/D3 cells (the intersection in the upper Venn diagram of Criterion I, Figure 5a). Furthermore, based on our recently reported expression profile of plasma transmembrane proteins in hCMEC/D3 cells (hereafter referred to as “BBB PM” [17]), we identified 17 proteins (the intersection in the lower Venn diagram of Criterion I, Figure 5a) that were included in the list of previously identified BBB PMs. These 17 proteins were identified as hCMEC/D3-selective endocytic cell-surface proteins in hCMEC/D3 cells (lower Venn diagram in Criterion I, Figure 5a). This list of 17 proteins includes ICAM1 (Figure 5b).

As we considered that the endocytic cell-surface proteins in hCMEC/D3 cells but not in HUVECs represent likely candidates for the delivery of biopharmaceuticals to the brain, we also extracted the endocytic cell-surface proteins found in hCMEC/D3 cells but not in HUVECs from among the 61 hCMEC/D3-selective “biotinylated endocytic cell-surface proteins” (Criterion II). As a result, 12 proteins were identified (upper Venn diagram in Criterion I, Figure 5a); among these, eight are included in the previously published list of BBB PMs. These eight proteins were identified as hCMEC/D3-selective endocytic cell-surface proteins (Criterion II, Figure 5a). This list of eight proteins includes TFRC (Figure 5b).

We also considered that the endocytic proteins that are more abundant in hCMEC/D3 cells than in HUVECs are likely candidates for the delivery of biopharmaceuticals to the brain. Therefore, we extracted the endocytic cell-surface proteins with higher peak areas in hCMEC/D3 cells than in HUVECs, using Criterion III. The mean fold change of the peak area ratio between hCMEC/D3 cells and HUVECs was 1.19. When the threshold fold change of the peak area ratio was set at 4, 14 proteins among the 64 “biotinylated endocytic cell-surface proteins” occurring in both hCMEC/D3 cells and HUVECs (the intersection in the Venn diagram, Appendix A) were extracted as hCMEC/D3-selective “biotinylated endocytic cell-surface proteins”, of which nine are included the previously published list of BBB PMs (Criterion III, Figure 5a).

A total of 34 proteins selected via criteria I–III were identified as hCMEC/D3-selective endocytic cell-surface proteins. The peak areas of the 34 identified endocytic cell-surface proteins in the “Internalization” fraction are shown in Figure 5b. ICAM1 (satisfying Criterion I), TFRC (satisfying Criterion II), and cell-surface hyaluronidase (CEMIP2) (satisfying Criterion III) exhibited the highest peak areas among the identified proteins selected using criteria I–III. The internalization efficiency of the 34 identified proteins was estimated according to the peak area ratio of “biotinylated endocytic cell-surface proteins”-to-“biotinylated cell-surface proteins” (Figure 5c). The peak area ratios of SYNGR2, CD97, CEMIP2, ITGA3, LAMP1, and CD58 were 0.5–1.0, suggesting that their internalization efficiency was higher than that of the other proteins. In contrast, the peak area ratio of ICAM1 was <0.1, suggesting that the internalization efficiency of ICAM1 was lower than that of the other proteins. The peak area ratios of LDLR, TFRC, and PODXL were >1 because these proteins were identified using Criteria B (Figure 3e), suggesting that the internalization efficiency of these proteins was overestimated.

### 3.4. Identification of Brain Microvessel Endothelial Cell-Selective Endocytic Cell-Surface Proteins

The expression and localization of the 34 identified proteins in human brain microvessels represent important aspects to consider in determining candidate cell-surface proteins for the delivery of biopharmaceuticals to the brain across the BBB. The open database of mRNA expression levels in human endothelial cells based on RNA-seq analysis (https://www.brainrnaseq.org/) [21], our previously published list of BBB PMs [17], and the human protein atlas, which provides protein localization information based on immunohistochemistry (https://www.proteinatlas.org/) [19,20] are available as resources to evaluate these issues. We therefore compared the peak areas of the 34 biotinylated endocytic proteins to their mRNA expression levels in human brain endothelial cells (Figure 6a). The mRNA expression of the 34 identified proteins, extracted from the RNA-seq database [21], is summarized in Appendix A. Among these, the mRNA expression of seven proteins in human brain endothelial cells was relatively higher, with fragments per kilobase of transcript per million mapped reads (FPKM) > 5 (whereas the median peak of FPKM was estimated to be almost 1) [21]. The levels of the seven extracted proteins in the plasma membrane fraction were almost the same in hCMEC/D3 cells as in HBMEC/Ciβ, another human brain microvascular endothelial cell line (Figure 6b). Data from the human protein atlas showed that five of these seven proteins (Podocalyxin (PODXL), TFRC, ICAM1, Integrin subunit alpha 1 (ITGA1), and CEMIP2) were expressed in brain microvessel endothelial cells (Figure 6c) [19,20]. The tissue distribution of these five identified proteins (Appendix A) reveals that none of the proteins had brain-specific expression in the body. The distribution of PODXL and ICAM1 in peripheral tissue was limited compared to that of TFRC and other proteins. These results suggest that PODXL and ICAM1 could be classified as likely BBB-selective endocytic cell-surface proteins for brain-targeting drug delivery systems.

### 3.5. Internalization of PODXL into hCMEC/D3 Cells

We focused on PODXL as a novel BBB-selective endocytosis protein in human brain microvessels, because *PODXL* mRNA expression in human brain endothelial cells was highest among the identified proteins (Figure 6a). Immunohistochemistry using an anti-PODXL antibody revealed that PODXL was predominantly expressed in the plasma membrane of hCMEC/D3 cells (Figure 7a). To validate the internalization of cell-surface PODXL into hCMEC/D3 cells, the internalization of PODXL upon treatment with 20% human serum was examined using an anti-PODXL antibody. FACS analysis using this antibody verifies the binding of cell-surface PODXL in intact cells. The internalization of FITC-labeled anti-PODXL antibodies was observed within 5 min (Figure 7c, white arrowheads), whereas FITC-labeled IgG was not detected (Figure 7b).

### 3.6. Immunohistochemical Analysis of PODXL in Human Cerebral Cortex

Analysis of the protein atlas data revealed that PODXL is predominantly localized in human brain endothelial cell [18,19]. To clarify whether PODXL is localized in human brain microvessels, and to identify the side of the microvessel membrane on which PODXL is localized, we conducted immunohistochemical analysis of PODXL localization in human cerebral cortex sections using the anti-PODXL antibody. This revealed that PODXL was expressed in brain microvessels; PODXL was identified staining with FL-labeled lectin (Figure 8a,b). The high-magnification images show that PODXL was colocalized with FL-labeled lectin, suggesting its localization at the luminal membrane of human microvessels (Figure 8c).

## 4. Discussion

In the present study, we have established an in vitro method for comprehensive identification of endocytic cell-surface proteins in cells via surface biotinylation methodology and SWATH-MS-based quantitative proteomics. Using this method, we identified a variety of endocytic proteins in hCMEC/D3 cells and HUVECs, including CD antigens, receptors, SLC transporters, and enzymes expressed on the plasma membrane. Plasma membrane proteins are internalized into the cells via several pathways such as clathrin- or caveolin-dependent pathways, and via macropinocytosis. In the present study, endocytic cell-surface proteins internalized via clathrin-dependent endocytosis such as TFRC, caveolin-dependent endocytic cell-surface proteins such as insulin-like growth factor 1 receptor (IGF1R) [26], clathrin- and caveolin-independent endocytic proteins such as ICAM1 [27], and macropinocytosis-mediated endocytic proteins such as ITGB1 [28], were identified. These findings suggest that our established method is useful to identify a spectrum of plasma membrane proteins whose internalization into cells is mediated by different internalization mechanisms.

MCAM (also known as CD146) and LGALS1 were identified as being internalized at relatively high levels in both hCMEC/D3 cells and HUVECs, relative to the other identified endocytic proteins. Data from the RNA-seq database [21] and the human protein atlas [19,20] suggest that MCAM (mRNA expression: 2.40 FPKM; protein expression level: high), and LGALS1 (mRNA expression: 16.9 FPKM; protein expression level: medium) are expressed and localize in human brain microvessels. MCAM is overexpressed in high-grade gliomas, and the majority of glioma stem cells expressing CD133 are also MCAM-positive [29]. In addition, the mRNA expression of LGALS1, which regulates the growth and metastasis of glioma cells [30,31], correlates with the malignant potential of human gliomas. These findings suggest that MCAM and LGALS1 represent therapeutic targets for glioblastoma. Moreover, our present findings further indicate that MCAM and LGALS1 also appear to constitute promising candidate molecules to deliver biopharmaceuticals from the blood across the BBB to glioblastomas.

hCMEC/D3 is an immortalized human brain microvascular endothelial cell line [22], whereas HUVECs are freshly isolated from pooled donors. Our quantitative proteomics study [17] shows the validity of using hCMEC/D3 cells as a human BBB model. Those findings suggest that hCMEC/D3 cells retain the protein expression levels of human brain microvessel transporter and receptor proteins. For in vitro study, Weksler et al. [32] recommended using hCMEC/D3 cells up to the 35^th^ passage, whereas we used passages 33–40. Although it has not been reporter that protein expression in hCMEC/D3 cells changes after the 35^th^ passage, this is a potential limitation of our in vitro study. To overcome this issue, we examined the protein expression of the proteins we identified in our study, using data from a study on another human BBB model cell, HBMEC/ciβ, determined using SWATH-MS [17]. Furthermore, using human brain RNA-seq database [21] and the human proteome atlas [19,20], we evaluated the mRNA and protein expressions of the identified proteins in this study. In summary, we identified the BBB-selective proteins, which do not occur in HUVECs.

For efficient delivery of biopharmaceuticals across the BBB, it is necessary to minimize internalization of biopharmaceuticals by peripheral microvascular endothelial cells for efficient delivery across the BBB. ICAM1 and PODXL were identified as BBB-selective endocytic cell-surface proteins by considering not only the difference in cell-surface and endocytic protein expression profiles between hCMEC/D3 cells and HUVECs, but also mRNA expression in human brain endothelial cells, plasma membrane protein expression in human brain microvascular cells, protein localization in human brain microvessels, and protein distribution in peripheral tissues. ICAM1 and PODXL were identified using different criteria. ICAM1 was identified as both a “biotinylated cell-surface protein” and “biotinylated endocytic protein” in hCMEC/D3 cells (Criterion I). PODXL was identified as a “biotinylated endocytic protein” only in hCMEC/D3 cells, and a “biotinylated cell-surface protein” in both hCMEC/D3 cells and HUVECs (Criterion II). Based on our criteria, ICAM1 showed greater selectivity for endocytosis in human brain microvascular endothelial cells than PODXL. The peak area of ICAM1 in the plasma membrane fraction of hCMEC/D3 cells and HBMEC/Ciβ was about 10-fold higher than that of PODXL, and was almost the same as that of TFRC. Therefore, these findings suggest that ICAM1 is a better candidate molecule for delivery of biopharmaceuticals to the brain than PODXL, due to its greater protein expression in the brain microvascular endothelial cells.

Among the internalized cell-surface proteins in hCMEC/D3 cells, ICAM1 has the highest expression level, as determined by Criterion I. It is expressed at the luminal side of human brain microvessels, as shown by immunohistochemistry [33,34]. ICAM1 promotes leukocyte adhesion and transcellular migration of vascular endothelium [35,36,37], as leukocytes are much larger than the ligands for receptors that are involved in clathrin-mediated endocytosis (vesicle diameter, −100 nm) or caveolar-mediated endocytosis (vesicle diameter, 50–80 nm). Consistent with this ability, nanocarriers (100–250-nm polymer particles) coated with anti-ICAM1 antibodies were internalized via cell adhesion molecule -mediated endocytosis and were transported by transcytosis across human gastrointestinal cell and brain microvascular endothelial cell monolayers without disrupting the permeability barrier [38,39,40]. In the plasma membrane of hCMEC/D3 (this study) and HBMEC/Ciβ cells [17], the expression level of ICAM1 is similar to that of TFRC. We estimated that the endocytic efficiency of ICAM1 was 0.0929, suggesting that the internalization rate of ICAM1 into human brain microvessels is low.

Among the internalized cell-surface proteins in hCMEC/D3 cells, ICAM1 has the highest expression level, as determined by Criterion I. It is expressed at the luminal side of human brain microvessels, as shown by immunohistochemistry [33,34]. ICAM1 promotes leukocyte adhesion and transcellular migration of vascular endothelium [35,36,37], as leukocytes are much larger than the ligands for receptors that are involved in clathrin-mediated endocytosis (vesicle diameter, –100 nm) or caveolar-mediated endocytosis (vesicle diameter, 50–80 nm). Consistent with this ability, nanocarriers (100–250-nm polymer particles) coated with anti-ICAM1 antibodies were internalized via cell adhesion molecule -mediated endocytosis and were transported by transcytosis across human gastrointestinal cell and brain microvascular endothelial cell monolayers without disrupting the permeability barrier [38,39,40]. In the plasma membrane of hCMEC/D3 (this study) and HBMEC/Ciβ cells [17], the expression level of ICAM1 is similar to that of TFRC. We estimated that the endocytic efficiency of ICAM1 was 0.0929, suggesting that the internalization rate of ICAM1 into human brain microvessels is low.

ICAM1 and ICAM2 are transmembrane glycoproteins containing five and two Ig domains, respectively. The two Ig domains of ICAM2 are homologous to the two amino-terminal domains of ICAM1. ICAM1 was internalized into hCMEC/D3 cells while ICAM2 was internalized in HUVECs. However, it is reported that the amino acid sequences of ICAM1 and ICAM2 had only 34% identity [41]. In addition, ICAM1 is upregulated by inflammatory cytokines, whereas ICAM2 is not [39,40,41]. Thus, the risk of cross-reaction between anti-ICAM1 antibody to ICAM2 seems to be low. By contrast, anti-ICAM1 antibody-labeled biopharmaceutical and nanocarrier delivery is unlikely to be brain-specific, because ICAM1 is expressed in the liver and kidney as well as in the brain. ICAM1 expression is upregulated at the BBB under inflammatory conditions and ischemia [42,43,44], and in CNS inflammatory diseases such as multiple sclerosis and experimental allergic encephalitis [45]. Furthermore, ICAM-1 expression is elevated in brain tumors [46]. Thus, ICAM1 may serve as a targetable cell-surface protein for the delivery of biopharmaceuticals under pathological conditions in CNS.

PODXL, a type I transmembrane protein with a short cytoplasmic domain and highly sialylated and glycosylated domains [47], supports E/L-selectin-mediated leukocyte adhesion in peripheral blood vessels [48]. We found that PODXL localized within the plasma membrane of hCMEC/D3 cells and the luminal membrane of human brain microvessels. Furthermore, among the 34 identified candidates, PODXL mRNA had the highest expression in isolated human brain endothelial cells. The expression of PODXL mRNA was higher in isolated bovine brain microvessels than in microvessels derived from the lung, liver, heart, and kidney, as determined by quantitative PCR profiling [49]. PODXL internalization was observed in hCMEC/D3 cells, but not in HUVECs. Therefore, PODXL may also constitute a novel candidate cell-surface protein for delivery of biopharmaceuticals to the brain.

TFRC is experimentally the first targeted RMT protein at the BBB [2]. We identified TFRC as a cell-surface protein selectively internalized through the BBB, rather than into peripheral microvascular endothelial cells. This result suggests that TFRC constitutes a target cell-surface protein at the BBB for the delivery of biopharmaceuticals to the brain. In contrast, INSR, LRP1, 4F2hc, and GLUT1, which are reported to be molecules targetable for transcytosis across the BBB [7,8,9,10], were not identified as internalized proteins in the present study, because these proteins did not satisfy our criteria. In particular, although INSR contains 45 primary amines that remain extracellular, it was not identified either in the plasma membrane or the surface fraction in the present study. Previously, we reported that the expression level of INSR was 0.720 fmol/μg protein in the plasma membrane fraction of hCMEC/D3 cells, using targeted quantitative proteomics [18]. As targeted quantitative proteomics is more sensitive than label-free quantitative proteomics, those findings suggest that the expression level of INSR in hCMEC/D3 cells was too low to be detected by label-free quantitative proteomics with SWATH-MS.

In turn, although 4F2hc and LRP1 were identified as biotinylated cell-surface proteins, their endocytic activity did not satisfy our criterion requiring stripping efficiency > 85% (72.2% and 34.5%, respectively). GLUT1 was identified in the plasma membrane fraction of hCMEC/D3 and HBMEC/ciβ cells [17], but not in the cell-surface fraction of hCMEC/D3 cells, which suggests that GLUT1 was not biotinylated. Biotinylation of sulfo-NHS-SS-biotin requires a primary amine, lysine, or N-terminus to be surface accessible. Although GLUT1 contains five extracellular primary amines, it is speculated that GLUT1 may have a three-dimensional structure wherein the sulfo-NHS-SS-biotin is hard to attack. These findings suggest a limitation of our current established method. It is reported that other cell-surface modification reagents such as aminooxy-biotin may allow identification of other plasma membrane proteins that were not modified by sulfo-NHS-SS-biotin [13]. Thus, improvement of our established method is necessary to comprehensively examine the internalization of cell-surface proteins into cells.

## 5. Conclusions

In conclusion, we have established an in vitro identification method for endocytic cell-surface proteins, through a combination of cell-surface biotinylation and SWATH-MS-based quantitative proteomics. Our present study demonstrates that six cell-surface proteins were selectively internalized into human brain microvascular endothelial cells, rather than peripheral microvascular endothelial cells. Further, using public databases, two cell-surface proteins were identified as BBB-selective endocytic cell-surface proteins. Strategies such as using a bispecific monoclonal antibody that can bind to ICAM1 or PODXL and to disease-targeting molecules, and using monospecific anti-ICAM1 or anti-PODXL antibodies or ligand-labeled biopharmaceuticals and nanocarriers nanoparticles, may provide effective targeted delivery to the brain across the BBB for the treatment of CNS diseases. Together, our findings provide novel strategies for the development of a BBB-permeable drug delivery system.

## Figures and Tables

**Figure 1 pharmaceutics-12-00579-f001:**
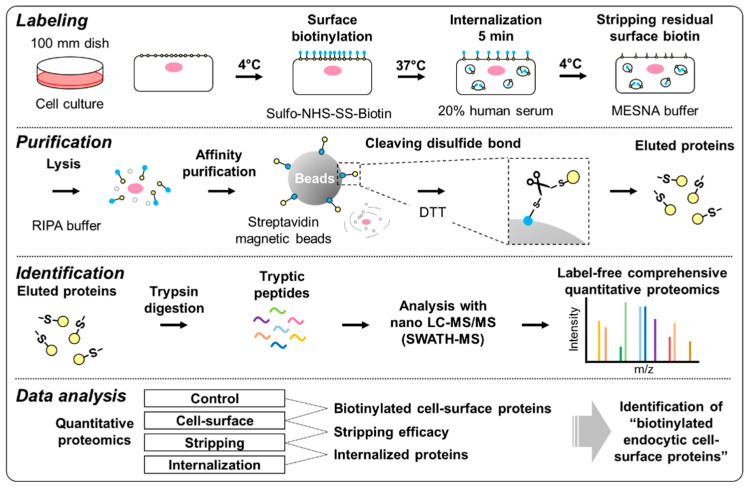
Experimental outline of the identification of “biotinylated endocytic cell-surface proteins” in the cells by a combination of cell-surface biotinylation methodology and SWATH-MS-based quantitative proteomics. Labeling: Cells were treated with sulfo-NHS-SS-Biotin at 4 °C for 30 min, then with 20% FBS at 37 °C for 5 min to allow protein internalization. Residual cell-surface proteins were removed by treatment with MESNA buffer. Purification: Following cell lysis with RIPA buffer, biotinylated proteins were collected using streptavidin magnetic beads. After washing the beads, the proteins were eluted from the beads by cleavage of the disulfide bonds of sulfo-NHS-SS-Biotin using DTT. Identification: The eluted proteins from streptavidin magnetic beads were digested with trypsin, then tryptic peptides were analyzed via SWATH-MS-based quantitative proteomics. Data analysis: Selection of “biotinylated cell-surface proteins” and “biotinylated endocytic cell-surface proteins” was performed as described in Section 3.2.

**Figure 2 pharmaceutics-12-00579-f002:**
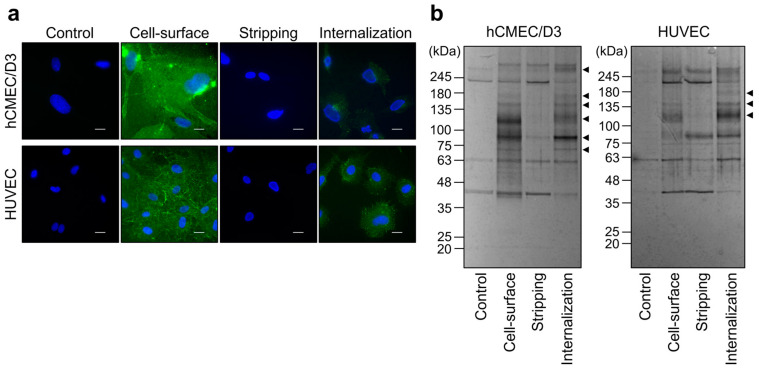
Internalization of biotinylated cell-surface proteins in hCMEC/D3 cells and human umbilical vein endothelial cells (HUVECs). (**a**) Representative fluorescence images of biotinylated proteins detected by FITC-labeled streptavidin (green) in hCMEC/D3 cells and HUVECs. The biotinylated proteins were stained with FITC-streptavidin (green), and the nuclei were stained with DAPI (blue). Fluorescence in the cells was observed using fluorescence microscopy (*n* = 3). Scale bar, 20 μm. (**b**) Representative silver staining images of biotinylated proteins eluted from streptavidin magnetic beads. Following solubilization of the cells, the biotinylated proteins from each fraction were recovered using streptavidin magnetic beads. The proteins bound to streptavidin magnetic beads were eluted by DTT, then subjected to SDS-PAGE and stained by silver staining (*n* = 3).

**Figure 3 pharmaceutics-12-00579-f003:**
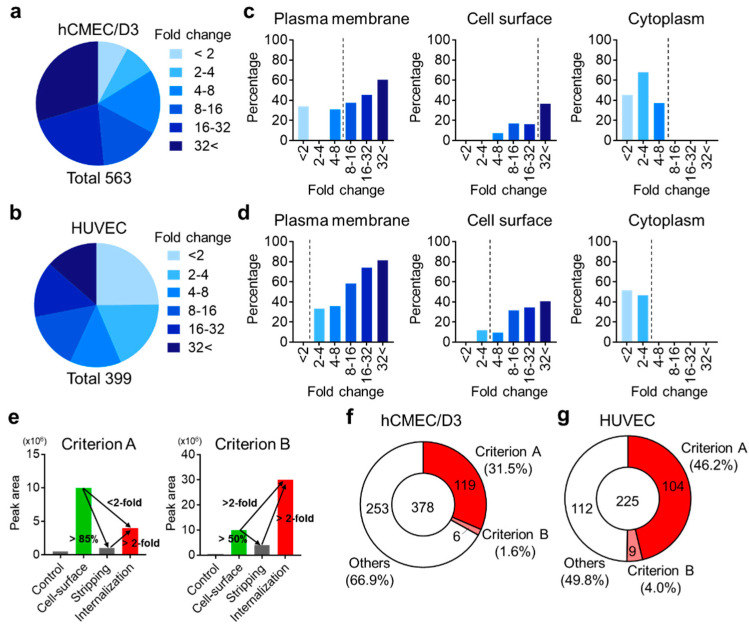
Identification of “biotinylated cell-surface proteins” and “biotinylated endocytic cell-surface proteins” in hCMEC/D3 cells and HUVECs. (**a**, **b**) “Cell-surface”-to-“Control” peak area ratio of all identified proteins in hCMEC/D3 cells (**a**) and HUVECs (**b**) were calculated, and the “Cell-surface”-to-“Control” peak area ratios of all identified proteins were grouped on a binary logarithmic scale. (**c**, **d**) The percentage of identified proteins annotated with “Plasma membrane”, “Cell surface”, or “Cytoplasm” in each group in hCMEC/D3 cells (**c**), and HUVECs (**d**) was calculated and grouped on a binary logarithmic scale. The threshold determined as indicating plasma membrane proteins was a “Cell-surface”-to-“Control” peak area ratio of 8 and 4, respectively, in hCMEC/D3 cells and HUVECs. (**e**) Scheme of criteria for the extraction of “biotinylated endocytic cell-surface proteins” in the cells (Criterion A and B). (**f**, **g**) Numbers of identified “biotinylated endocytic cell-surface proteins” in hCMEC/D3 cells (**f**) and HUVECs (**g**).

**Figure 4 pharmaceutics-12-00579-f004:**
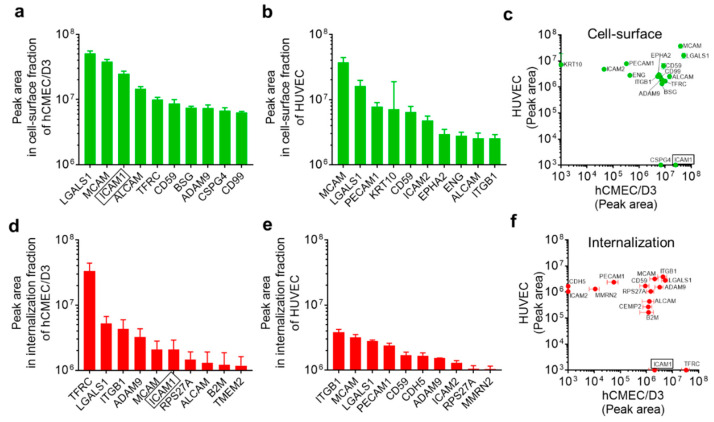
The top 10 highest expressing “biotinylated cell-surface proteins” and “biotinylated endocytic cell-surface proteins” in hCMEC/D3 cells and HUVECs. (**a**, **b**) Top 10 identified “biotinylated cell-surface proteins” in hCMEC/D3 cells (**a**) and HUVECs (**b**). (**c**) A total of 16 “biotinylated cell-surface proteins” were identified from among the top 10 identified “biotinylated cell-surface proteins” in either hCMEC/D3 cells or HUVECs. The relationship between the peak areas of the 16 “biotinylated cell-surface proteins” in hCMEC/D3 cells and HUVECs is shown. (**d**, **e**) Top 10 identified “biotinylated endocytic cell-surface proteins” in hCMEC/D3 cells (**d**) and HUVECs (**e**). (**f**) A total of 15 “biotinylated endocytic cell-surface proteins” were identified from among the Top 10 identified “biotinylated endocytic cell-surface proteins” in either hCMEC/D3 cells or HUVECs. The relationship between the peak areas of the 15 “biotinylated endocytic cell-surface proteins” in hCMEC/D3 cells and HUVECs is shown. Each datapoint represents the means ± SD (*n* = 3).

**Figure 5 pharmaceutics-12-00579-f005:**
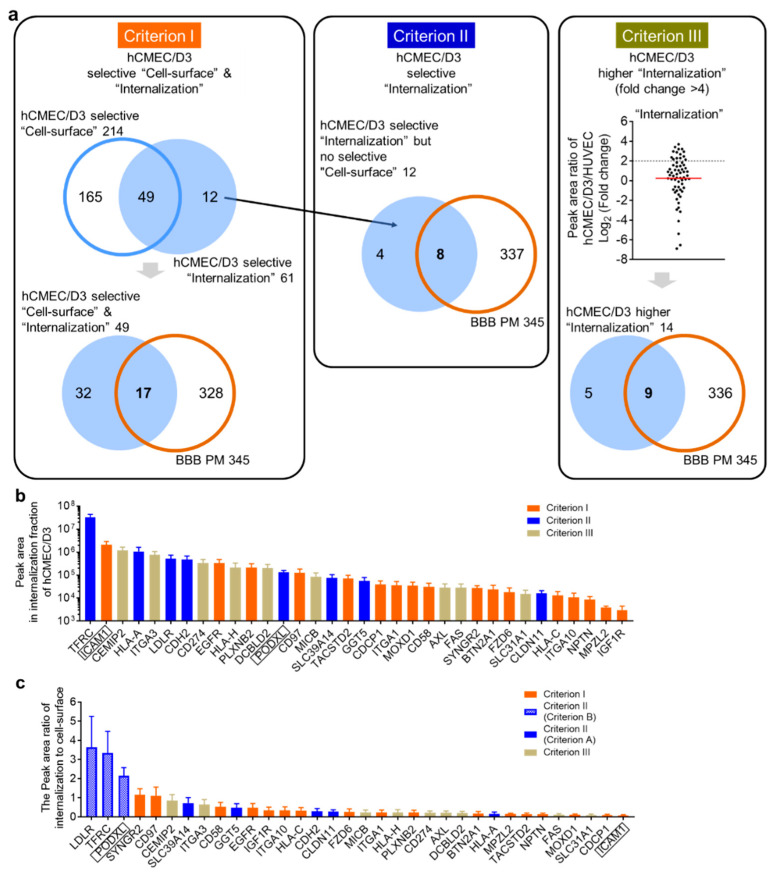
Identification of biotinylated endocytic cell-surface proteins in hCMEC/D3 cells compared to those in HUVECs. (**a**) Criterion I: Extraction of the internalized proteins from among the identified “biotinylated cell-surface proteins” and “biotinylated endocytic proteins” of hCMEC/D3 cells, but excluding those identified in HUVECs. Criterion II: Extraction of internalized proteins from among the identified “biotinylated endocytic proteins” of hCMEC/D3 cells, including only those identified in the cell surface of both hCMEC/D3 cells and HUVECs. Criterion III: Extraction of the proteins that were more highly internalized in the cell-surfaces of hCMEC/D3 cells than in those of HUVECs. BBB PM: plasma transmembrane protein expression in hCMEC/D3 cells, from our previous paper [17]. (**b**) Peak areas of the 34 identified biotinylated cell-surface proteins selectively internalized into the hCMEC/D3 cells. (**c**) Peak area ratios of “biotinylated endocytic cell-surface proteins” to “biotinylated cell-surface proteins” among the 34 identified proteins of hCMEC/D3 cells. Each bar represents the mean ± SD (*n* = 3).

**Figure 6 pharmaceutics-12-00579-f006:**
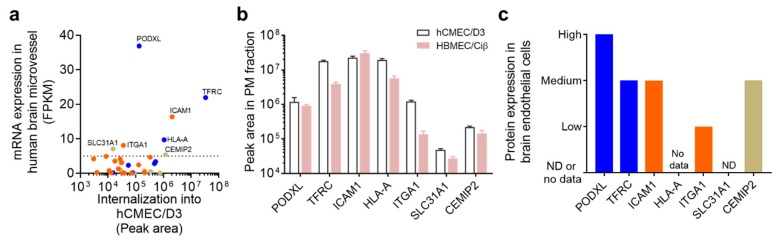
Identification of blood–brain barrier (BBB)-selective biotinylated endocytic cell-surface proteins in human brain microvascular endothelial cells. (**a**) Relationship between the expression of biotinylated endocytic cell-surface proteins in the hCMEC/D3 cells (peak area) and that of mRNA (FPKM) in human endothelial cells; the mRNA data are from the open database of mRNA expression levels in human endothelial cells [21]. (**b**) Comparison between the peak areas of seven identified proteins in hCMEC/D3 and HBMEC/Ciβ cells. These data are from our previously published paper [17]. Each datapoint represents the means ± SD (*n* = 3). (**c**) Localization of human brain endothelial cells by immunohistochemistry. These data are from the human protein atlas [19,20]. ND, not detected.

**Figure 7 pharmaceutics-12-00579-f007:**
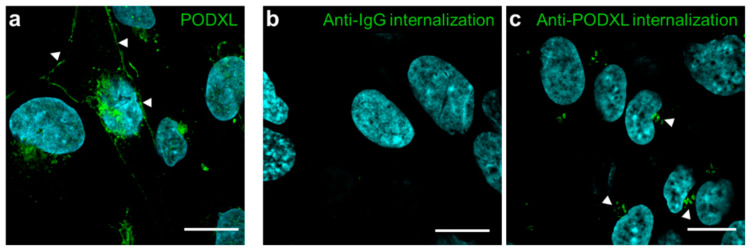
Localization and internalization of cell-surface PODXL into hCMEC/D3 cells. (**a**) Immunohistochemical analysis of PODXL in hCMEC/D3 cells was performed using an anti-PODXL antibody (green). Nuclei were stained using DAPI (blue). The images were obtained by confocal microscopy. Arrow heads show PODXL in the plasma membrane. (**b**, **c**) Internalization of cell-surface PODXL into hCMEC/D3 cells. The cells were treated with fluorescein (FL)-labeled anti-IgG (green, **b**) or FL-labeled anti-PODXL antibody (green, **c**) or FL-labeled anti-PODXL antibody (green, **c**) on ice for 30 min and then incubated at 37 °C for 5 min. Arrow heads show the internalized PODXL. Nuclei were stained using DAPI (blue). Scale bars: 20 μm.

**Figure 8 pharmaceutics-12-00579-f008:**
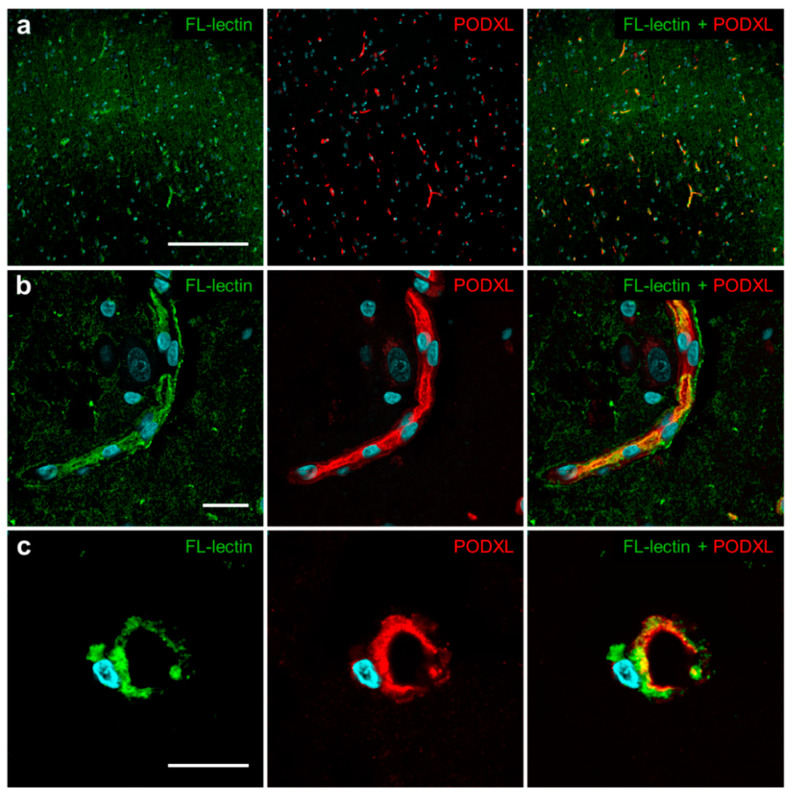
Localization of PODXL in the human cerebral cortex by confocal microscopy. Immunohistochemical analysis of PODXL was performed using an anti-PODXL antibody (red). Blood vessels were stained with fluorescein (FL)-conjugated lectin (green), and nuclei were stained using DAPI (blue). The images were obtained by confocal microscopy. The three images were obtained from different regions of the sections. Scale bars: 200 μm (**a**), 20 μm (**b**), 10 μm (**c**).

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
