# Peer review of "Identification of Cell-Surface Proteins Endocytosed by Human Brain Microvascular Endothelial Cells In Vitro"

_pharmaceutics, 2020, doi:10.3390/pharmaceutics12060579_

Round 1
Reviewer 1 Report
This manuscript by Ito, Oishi et al. reports an original approach to identify proteins localized at the surface of human brain endothelial cells and capable of undergoing endocytosis, which makes them interesting candidates to trigger BBB crossing. Using a combination of surface biotinylation followed by SWATH-MS-based quantitative proteomics, as well as immunocytochemistry, the authors identified some proteins with great potential in biopharmaceutical drug delivery to the brain, some of which were unreported as such up to now. Overall, this manuscript is well-written, well-structured, methods are sound and well-described and results adequately commented. I therefore recommend publication after a few points are addressed.
Important points to clarify:
1- The authors used two human immortalized cell lines to perform their study. However, they did not comment on how the immortalization process may have affected the general pattern of protein expression in those endothelial cells. Also, I noted that hCMEC/D3 cells (origin not detailed; please specify) were used between passages 33 to 40, which could moreover contribute to a modified protein pattern. As for HUVEC cells, passages were not precised; please specify and comment if relevant.
2- I noticed that 2 different confocal microscopes were used in this study. For reliability concerns, I would recommend detailing the conditions in which each of these microscopes were used (laser, stacks, illumination conditions, softwares for data acquisition and image processing, etc). In case this information makes the text too cumbersome, it could be placed in the Supplementary Information section.
3- Figure 7 (d,e): I have trouble interpreting those pictures by myself. Providing corresponding bright-field images would definitively improve the clarity and allow better understanding of the authors' own interpretation. Which portion of figure 7(d) is enlarged in 7(e) ?
Lines 431-433: High-magnification images showed that PODXL appears to be localized to the inside of the nucleus in brain microvessel endothelial cells, and showed colocalization with FITC-lectin, suggesting its localization at the luminal membrane of human microvessels (Figure 7(e)). How can PODXL be, at the same time, localized in the nucleus (which I can't personnally see) and at the luminal side of the microvesssel ? Is this image representative of all endothelial cells ? Could the authors provide some more images in SI ?
4- The authors compared the level of expression of PODXL in hCMEC/D3 and HUVEC cells. They concluded that 'Thus, although cell-surface expression of PODXL was observed in both hCMEC/D3 cells and HUVECs, the overall expression of PODXL is likely to be higher in the 501 former.' (lines 500-502) Would it not be possible to ascertain this assumption by comparing, in a statistical way, the normalized ratios of this endocytic protein levels in hCMEC/D3 and HUVEC cells ?
5- Finally, ICAM-1 was, unsurprisingly, identified as a BBB-selective endocytic cell-surface protein, displaying a low endocytic efficiency (lines 473-489). In Figure 4, the authors identified ICAM-2 as a major endocytic protein in HUVEC cells. Knowing that ICAM-1 and -2 are integrin-binding Ig superfamily adhesion molecules with close structural organization, what do the authors foresee of the use of ICAM-1 targeted biotherapy for brain delivery ? Would it really be brain-specific ?
Minor points to consider:
- I couldn't find Figure S2 cited in the text. Please enlarge Fig. S1 and S2.
- line 156: reference to formula ?
- lines 192-193: please specify what kind of replicates were used.
- lines 244-245: please elaborate a little more for non specialized readers.
- lines 253-254: are they the 'missing' 60 proteins for hCMEC/D3 and 61 proteins for HUVECS in Table S1?
- lines 336-344: please add some details, to help a better understanding. For instance, “biotinylated cell-surface proteins” (Figure 5 a, empty blue circle)
- some minor English mispelling and syntax error
Reviewer 2 Report
RE: Pharmaceutics-805541
The manuscript titled “Identification of cell-surface proteins endocytosed by human brain microvascular endothelial cells in vitro” submitted by Ito et al. introduces a new method for identifying the nature of internalized proteins. The paper takes a particular look at how endothelial cells of the BBB (represented by hCMEC/D3) differ in their internalization of surface proteins compared with systemic vascular cells (HUVEC). In the latter part of the paper, the researchers highlight the particular importance of several hCMEC/D3-specific internalized proteins such as ICAM-1 and PODXL, and offer a detailed functional explanation within the discussion. Finally, the paper also expounds upon the impacts on other proteins which fit important physiological roles but did not meet particular criteria (such as GLUT1). This study appears to be well controlled, with a clear hypothesis, significant data, and polished figures. However, minor stylistic issues exist, largely with the data figures.
- Figure 5a appears to restate a lot of the same information (just analyzed in a slightly different way) as Figure 3e-g.
- In figures 4 and 5, it would be helpful to put a box around ICAM-1 and PODXL, to illustrate their importance, as well as to remind readers that while both are of interest, they fall into different selection criteria.
- Figure 7 is somewhat confusing. 7c does not appear to actually show internalization, only staining on the surface for PODXL. Further, the latter part of the figure, which describes PODXL within the cerebral cortex, is not well-introduced in the text, and there are no stains for other proteins which are specifically not internalized, to be used as a control.
Round 2
Reviewer 2 Report
Authors responded to reviewers comments.